# Spatiotemporal evolution characteristics of coordinated development of urbanization and ecological environment in eastern Russia —Perspectives from the 3D global trend and 2D plane analysis

**Nan-Chen Chu**[1], **Xiang-Li Wu**[1]*, **Ping-Yu Zhang**[2,3]*

**1** College of Geographical Sciences, Harbin Normal University, Harbin, China, **2** Northeast Institute of Geography and Agroecology, Chinese Academy Sciences, Changchun, China, **3** University of Chinese Academy of Sciences, Beijing, China

* jndxwxl@163.com (XLW); zhangpy@iga.ac.cn (PYZ)

**Data Availability Statement:** All relevant data are within the paper.

## Abstract

Under the background of "the Belt and Road" and "the economic corridor of China, Mongolia and Russia" initiatives, it has great value to study the temporal and spatial evolution characteristics of the coordinated development between the urbanization and ecological environment in eastern Russia (the Siberian Federal District and the Far East Federal District). In this paper, we studied the urbanization development level, eco-environment development level, and their coupling coordinated development degree during 2005–2018 in the eastern Russia from the perspectives of the 3D global trend and 2D plane analysis. First, combining with the Population-Economic-Sociology and Pressure-State-Response models, the urbanization development level and eco-environment development level were calculated by the comprehensive weighting method of entropy weight and variation coefficient for eastern Russia. Second, the coupling coordinated development degree of the urbanization development level and eco-environment development level was measured by the coupling coordination model for eastern Russia. Finally, the spatial differentiation of the urbanization development level, the eco-environment development level and their coupling coordinated development degree was performed respectively by the 3D global trend and 2D plane analysis using ArcGIS. The results are as following. First, the comprehensive urbanization development level of eastern Russia has increased from 2005 to 2018, and the economic urbanization is the main factor that affects the urbanization development in eastern Russia. The comprehensive eco-environment development level of eastern Russia has decreased from 2005 to 2018, and the eco-environment pressure is the main factor that affects the eco-environment development in eastern Russia. The coupling coordination degree of the urbanization development and eco-environment development has increased from 2005 to 2018. However, it is still in the uncoordinated stage. Second, from 2005 to 2018, the urbanization development level of the Siberian Federal District is higher than that of the Far East Federal District. The eco-environment development level of the Siberian Federal District is

**Funding:** This work is supported by the National Natural Science Foundation of China (No. 42101165; 42071162), Heilongjiang Philosophy and Social Sciences Research Planning Project (No. 21JLC201), China Postdoctoral Science Foundation (No. 2021M693817), and Heilongjiang Postdoctoral Science Foundation (No. LBH-Z21067). Author Contributions National Natural Science Foundation of China(No. 42101165) Recipient: Nan-Chen Chu National Natural Science Foundation of China(No. 42071162) Recipient: Ping-Yu Zhang Heilongjiang Philosophy and Social Sciences Research Planning Project (No. 21JLC201) Recipient: Nan-Chen Chu China Postdoctoral Science Foundation (No. 2021M693817) Recipient: Nan-Chen Chu Heilongjiang Postdoctoral Science Foundation (No. LBH-Z21067) Recipient: Nan-Chen Chu The funders have roles in study design, data collection and analysis, decision to publish, or preparation of the manuscript.

**Competing interests:** The authors have declared that no competing interests exist.

balanced to that of the Far East Federal District. The coupling coordination degree of the Siberian Federal District is higher than that of the Far East Federal District. Among the Siberian and Far East Federal Districts, most of the federal subjects belong to the uncoordinated stage of the urbanization development and the eco-environment development. Third, the urbanization development level, the eco-environment development level, and their coupling coordinated development level are all spatially imbalanced in the eastern Russia, which show the "High West, Low East" and "High Center, Low North and Low South" spatial pattern from the perspectives of the 3D global trend and 2D plane analysis. The areas with high levels are concentrated in the Novosibirsk Region, Altay Territory, Kemerovo Region, Krasnoyarsk Territory, and Irkutsk Region. The areas with low ones are mostly in the Republic of Altay and Chukotka Autonomous Area. Finally, we suggest policies and strategies that can boost the growth and development of the urbanization and the eco-environment in the Sino-Russian border areas.

## Introduction

In 2013, China proposed "the Belt and Road" initiative, including "the Silk Road Economic Belt" initiative and "the 21st-Century Maritime Silk Road" initiative. In 2015, China issued the "Vision and Actions on Jointly Building Silk Road Economic Belt and 21st-Century Maritime Silk Road" initiative, which meant that "the Belt and Road" initiative came into a comprehensive promotion stage [1]. Among "the Belt and Road" initiative, "China- Mongolia- Russia economic corridor" initiative is constructed depending on the land transportation infrastructure [2]. Along China- Mongolia- Russia economic corridor, eastern Russia is an important neighbor area of Northeast China. As the key areas of "the Belt and Road" and "China- Mongolia-Russia economic corridor" initiatives, eastern Russia and Northeast China have came into the comprehensive opening-up and cooperation stage [2]. The coordinated development of urbanization and ecological environment is not only the global economic and social development trend, but also the important content of transnational economic cooperation between China and Russia.

Research on the urbanization and eco-environmental development in eastern Russia concentrates on the urbanization development process analysis, the urbanization development problems discussion, and the different eco-environmental problems research. (1) Urbanization development process analysis. Shang analyzed that the urbanization of eastern Russia had experienced three important stages [3], including the early eastward expansion of the Moscow Principality, the Russian expansion in the Amur River Basin, and the construction climax in the Tsarist Russian period. Relying on the rapid development of industrialization, the urbanization development of eastern Russia went through with military and administrative functions, which was from west to east developing, from the center to around radiating [4–6]. Mishchuk reviewed the urbanization immigrants characteristics in the Far East Federal District [7], including the European immigrants settling in the Far East Federal District, the forced and voluntary immigrants to the Far East Federal District, and the reverse migration flow formation in the Far East Federal District. (2) Urbanization development problems discussion. The urbanization development was rapidly promoted by the industrialization development in the Soviet Union. After the disintegration of the Soviet Union, Russia came into the transition process from the planned economy to the market economy. Russia's urbanization was in the

stagnation process because of the radical economic transformation and the regional serious crisis. The urbanization development was characterized by the shrinking and extinction of small cities, the negative growth of urban and rural population, the unstable development of urban system [8]. Gao and Li thought that the urban construction was developing slowly in the eastern Russia from 2010. There were a series of problems in the urbanization development, such as the slow urban population agglomeration, the small number of mega cities, the single industrial structure of small and medium-sized cities, the urban imperfect hierarchical scale structure, the uneven distribution of urban spatial structure [5, 9]. Kuleshov believed that the interaction between eastern Russia and Northeast China would solve the regional development imbalance and internal urbanization problems in eastern Russia [10]. (3) Different eco-environmental problems research. While developing mining, forest harvesting and high-consuming military enterprises, the eco-environmental problems such as the air pollution, water pollution, radioactive pollution, soil heavy metals, offshore pollution and forest resources loss became more and more serious in eastern Russia [11, 12]. Bityukova believed that the urban air pollution severity mainly depended on the specialization degree of the industrial sectors in eastern Russia [13]. Vasilenko thought that the depletion and deterioration of water resources were mainly affected by human water resources activities in the southern areas of western Siberian Federal District [14]. Zhuravel analyzed the impact of coal mining on the ecological environment. He put forward the suggestions on sustainable and optimal operation efficiency in the southern Yakutia and Kuznetsk basins [15]. Based on the comprehensive index of human environmental intensity, Bityukova evaluated the urban eco-environment status, including the air and water pollution, solid waste quantity, thermal pollution and radiation pollution in eastern Russia [16]. At present, research on the urbanization and eco-environment concentrates on unilateral analysis, there are relatively few studies on the coordinated development of urbanization and eco-environment in eastern Russia. Moreover, on the basis of long-time series of historical documents and historical data, Chinese scholars have carried out the qualitative research, judgment and discussion on urbanization and eco-environment in eastern Russia. However, due to the data limitations, the quantitative and visual research is obviously insufficient on the urbanization and eco-environment in eastern Russia. Especially in the past 10 years, research on the coordinated development of urbanization and eco-environment needs to be continuously updated in eastern Russia.

Eastern Russia contains twenty-one federal subjects. Among these federal subjects, the physical geography, energy resources, political affairs, economic level and social level are all different. After the disintegration of the Soviet Union, Eastern Russia has experienced severe economic crises. Its urbanization and eco-environment are still in the uncoordinated development condition. Eastern Russia is an important neighbor area of Northeast China. However, it is not clear for China to know the urbanization and eco-environment development conditions of eastern Russia. This phenomenon has seriously affected the cross-border cooperation in economy, trade, industry, transportation, communication, resources and environment between China and Russia. By studying the urbanization and eco-environment development patterns of eastern Russia, it would be helpful to select cross-border cooperation nodes, cooperation channels and cooperation platforms for China and Russia. It also would be helpful to explore the complementary growth points of regional cooperation, so as to form a coordinated development pattern for China and Russia. Therefore, it is very important to study the coordinated development pattern of urbanization and eco-environment in eastern Russia. In this paper, we studied the urbanization development level, eco-environment development level, and their coupling coordinated development degree during 2005–2018 in the eastern Russia from the perspectives of the 3D global trend and 2D plane analysis. First, combining with the Population-Economic-Sociology and Pressure-State-Response models, the urbanization index

system and the eco-environment index system were constructed for eastern Russia. Second, based on the comprehensive weighting method of entropy weight method and variation coefficient weight method, the urbanization development level and eco-environment development level were calculated for eastern Russia. Third, the coupling coordinated development degree of the urbanization development level and eco-environment development level was measured by the coupling coordination model for eastern Russia. Finally, the spatial differentiation of the urbanization development level, the eco-environment development level, and their coupling coordinated development degree was performed respectively by the 3D global trend and 2D plane analysis in ArcGIS. In the theory, this research could enrich the theoretical systems of urban geography and urban ecology. It could provide guidance and reference for similar related research in other parts of the world, especially in China's adjacent areas. In the practice, this research could clear the complementary points of bilateral cooperation for China and Russia. It could provide scientific reference for regional development planning, economic optimization layout, energy and resource development, infrastructure construction for the adjacent areas of China and Russia. It could provide suggestions for expanding the economic cooperation field and the economic investment scale for the border cities of China and Russia. It also could provide policy implications for the border trade, transportation facilities, border tourism, border cooperation zone, eco-environment protection of China and Russia.

## Materials and method

### Study area

Eastern Russia contains the Siberian Federal District and the Far East Federal District (Fig 1), which connects the Asia Pacific region and the European region. It is the frontier region of Russia's foreign economic cooperation. Siberian Federal District contains ten federal subjects, including Republic of Altay, Republic of Tyva, Republic of Khakasia, Altay Territory, Krasnoyarsk Territory, Irkutsk Region, Kemerovo Region, Novosibirsk Region, Omsk Region and Tomsk Region (Fig 1). It has an administrative area of 4.36 million square kilometers, a population of 17.17 million, a population density of 3.9 people per square kilometer, a population urbanization rate of 74%, a regional GDP of 7.76 trillion rubles in 2018. Far East Federal District contains eleven federal subjects, including Republic of Buryatia, Republic of Sakha(Yakutia), Zabaikalsk Territory, Kamchatka Territory, Primorsky Territory, Khabarovsk Territory, Amur Region, Magadan Region, Sakhalin Region, Jewish Autonomous Area and Chukotka Autonomous Area (Fig 1). It has an administrative area of 6.95 million square kilometers, a population of 8.19 million, a population density of 1.2 people per square kilometer, a population urbanization rate of 73%, and a regional GDP of 3.88 trillion rubles in 2018. Due to the financial resources lack for maintaining the normal operation of eco-environment infrastructures, it is difficult for the government to modernize traditional environmental facilities and popularize the high-tech pollution prevention technologies. Some factories discharge large amounts of waste and toxic gases into the air. The urbanization quality have become relatively low and the eco-environment problems have become increasingly serious in eastern Russia.

### Research methods

**Index systems of urbanization and eco-environment.** In the process of urbanization development, people is the subject of behavior capacity and the eco-environment is the subject of carrying capacity. The agglomeration of economic and industrial factors causes stress and squeeze effects on the eco-environment. At the same time, with the application and popularization of high and new technologies, the urban resource intensive utilization and environmental pollution control have been improving. With the improvement of eco-environment

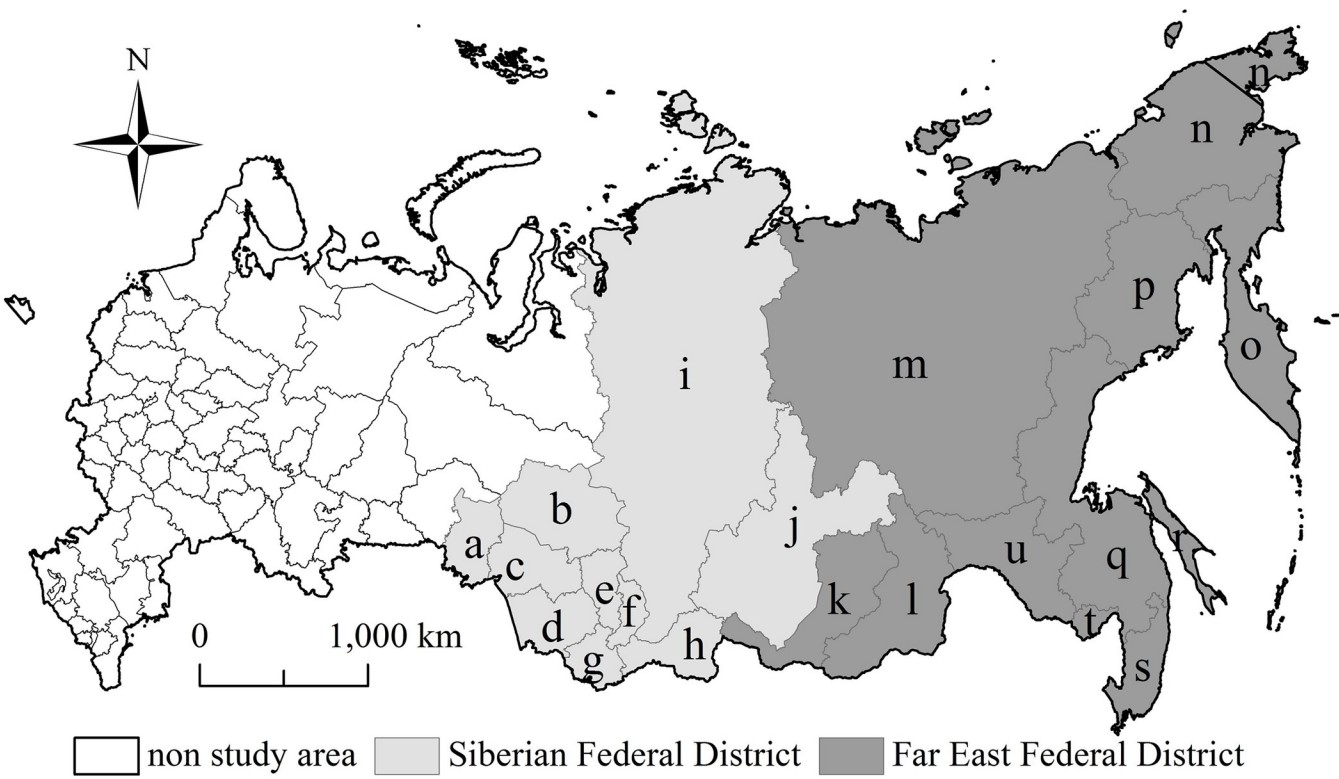

**Fig 1. Sketch map of the study area.** Note: a Omsk Region. b Tomsk Region. c Novosibirsk Region. d Altay Territory. e Kemerovo Region. f Republic of Khakasia. g Republic of Altay. h Republic of Tyva. i Krasnoyarsk Territory. j Irkutsk Region. k Republic of Buryatia. l Zabaikalsk Territory. m Republic of Sakha (Yakutia). n Chukotka Autonomous Area. o Kamchatka Territory. p Magadan Region. q Khabarovsk Territory. r Sakhalin Region. s Primorsky Territory. t Jewish Autonomous Area. u Amur Region. This drawing has not been previously copyrighted. The authors created the image themselves.

carrying capacity, cities become more and more suitable for human habitation. Therefore, the urbanization development could bring "destruction" effect or "improvement" effect to the eco-environment, the eco-environment could also produce "containment" effect or "promotion" effect on urbanization development [17, 18]. According to the references [19–28], in order to accurately evaluate the urbanization development level, ecological environment development level and their coordinated development degree, the urbanization evaluation index system and eco-environment evaluation index system are constructed by following the principles of scientificity, representativeness, feasibility, comparability, systematicness and data availability in eastern Russia, as shown in Tables 1 and 2.

*(1) Urbanization evaluation system based on Population-Economic-Sociology (PES) model.* Urbanization is the economic and social development process in which non-agricultural industries gather in the cities, rural population migrates to non-agricultural industries, the number of cities increases, the scale of cities expands, the mode of production and lifestyle of cities spread to rural areas, the material and spiritual civilization of cities spread to rural areas [19]. Based on the Population-Economic-Sociology (PES) model, we construct the urbanization evaluation index system of eastern Russia [18–24] (Table 1). The urbanization development levels with regard to the population urbanization, economic urbanization and social urbanization aspects are measured combining with fourteen indicators. Among them, population urbanization is the important content of urbanization development for eastern Russia, which is the process of rural population transferring to urban population. Economic urbanization refers to the aggregation of economic factors and non-agricultural industries to cities, so

**Table 1. Evaluation system of the urbanization in eastern Russia.**

| First-Level Indices | Second-Level Indices | Third-Level Indices, Unit | Attribute | Entropy weight | Variation weight | Comprehensive weight |
|---|---|---|---|---|---|---|
| Urbanization system | Population urbanization | Proportion of non-agricultural population, % | + | 0.01 | 0.02 | 0.01 |
| | | Population size (year-end), thousands | + | 0.09 | 0.09 | 0.09 |
| | | Population density, person/km$^2$ | + | 0.16 | 0.13 | 0.15 |
| | | Annual average number of employees, thousands | + | 0.09 | 0.09 | 0.09 |
| | Economic urbanization | Proportion of non-agricultural industrial output value in GDP, % | + | 0.00 | 0.01 | 0.003 |
| | | Regional GDP, million rubles | + | 0.12 | 0.11 | 0.12 |
| | | Integrated budget revenue, million rubles | + | 0.08 | 0.09 | 0.09 |
| | | Fixed capital investment (in real terms), million ruble | + | 0.12 | 0.11 | 0.12 |
| | | Total retail trade volume (current price), million rubles | + | 0.11 | 0.10 | 0.11 |
| | Social urbanization | Highway operating mileage (at the end of the year), km | + | 0.10 | 0.10 | 0.10 |
| | | Number of hospital beds per 10000 population | + | 0.01 | 0.03 | 0.01 |
| | | Number of mobile radiotelephone connected user equipment per 1000 people, PCs | + | 0.02 | 0.03 | 0.02 |
| | | Number of students in general education institutions, thousands | + | 0.08 | 0.08 | 0.08 |
| | | Urban per capita residential housing area, m$^2$ | + | 0.00 | 0.02 | 0.01 |

as to promote economic development and enhance economic strength. Social urbanization is the improvement of social services, social infrastructure and life quality.

*(2) Eco-environment evaluation system based on Pressure-State-Response (PSR) model.* Eco-environment is the compound system of water resources, land resources and biological resources, which could affect the human survival and sustainable development. Based on the Pressure-State-Response (PSR) model, we construct the eco-environment evaluation index system of eastern Russia [20–28] (Table 2). The eco-environment development levels with regard to the eco-environment pressure, eco-environment state and eco-environment response aspects are measured combining with twelve indicators. Among them, eco-

**Table 2. Evaluation system of the eco-environment in eastern Russia.**

| First-Level Indices | Second-Level Indices | Third-Level Indices, Unit | Attribute | Entropy weight | Variation weight | Comprehensive weight |
|---|---|---|---|---|---|---|
| Ecological environment system | Ecological environment pressure | Pollutants discharged into the air from fixed sources, thousand tons | - | 0.11 | 0.11 | 0.11 |
| | | Polluted waste water discharged to surface water, million m$^3$ | - | 0.09 | 0.09 | 0.09 |
| | | Area of forest fire, hm$^2$ | - | 0.17 | 0.15 | 0.16 |
| | Ecological environment state | Sown area of crops, thousand hm$^2$ | + | 0.11 | 0.10 | 0.10 |
| | | Total grain harvest, thousand tons | + | 0.13 | 0.10 | 0.12 |
| | | Forest coverage, % | + | 0.01 | 0.02 | 0.01 |
| | | Area of forest land, thousand hm$^2$ | + | 0.08 | 0.08 | 0.08 |
| | | Supply of fresh water, million m$^3$ | + | 0.07 | 0.08 | 0.08 |
| | Ecological environmental response | Share of captured and neutralized air pollutants in total waste pollutants in fixed pollution sources, % | + | 0.01 | 0.02 | 0.01 |
| | | Recycled water capacity, million m$^3$ | + | 0.06 | 0.07 | 0.06 |
| | | Amount of pollutants collected from fixed sources, thousand tons | + | 0.10 | 0.10 | 0.10 |
| | | Reforestation area, thousand hm$^2$ | + | 0.08 | 0.08 | 0.08 |

environment pressure refers to the destruction, load and negative impact of human economic activities on the eco-environment. Eco-environment state is the health and safe condition of natural resources and ecosystem. Eco-environmental response refers to the feedback and resistance of the ecosystem in the face of human destruction, and it also refers to the measures taken by human beings when facing the security problems in the ecosystem.

**Index preprocessing.** Based on the spatial scale of the eastern Russian federal subjects, we construct the 21*14 urbanization original data matrix and the 21*12 eco-environment original data matrix during 2005–2018. Since the quantitative dimensions of 14 urbanization indicators and 12 eco-environment indicators of 21 federal subjects are quite different, it is necessary to standardize the original indicators and eliminate their quantitative dimensions. In this paper, the range standardization method is selected to remove the quantitative dimension of each index, so as to calculate the urbanization and eco-environment development levels of 21 federal subjects. The formula is as follows:

$$X_{ij} = \frac{x_{ij} - x_{j\min}}{x_{j\max} - x_{j\min}} \ (\text{``} + \text{''} \ \text{indicator}) \qquad X_{ij} = \frac{x_{j\max} - x_{ij}}{x_{j\max} - x_{j\min}} \ (\text{``} - \text{''} \ \text{indicator}) \tag{1}$$

where $x_{ij}$ is the original $j$ index value of the $i$ federal subject. $X_{ij}$ is the standardized $j$ index value of the $i$ federal subject. $x_{j\max}$ is the maximum value of the $j$ index. $x_{j\min}$ is the minimum value of the $j$ index.

**Index weight determination.** The methods for determining the indicators' weight contain subjective weighting method and objective weighting method. Since the subjective weighting method is tendentious and the single objective weighting method has some errors, the two objective weighting methods of entropy weighting and coefficient variation weighting are selected to calculate the urbanization development level and the eco-environment development level of 21 federal subjects. The combination of entropy weighting and coefficient variation weighting could not only avoid the uncertainty of the subjective weighting method, but also could get rid of the defects of the single objective weighting method [29].

*(1) Entropy weighting method.* Entropy is used to measure the order or disorder degree of a system in classical physical thermodynamics. In this paper, we draw lessons from the concept of entropy weight, characterizing the possible state of various indicators and their contribution to the urbanization system or eco-environment system. The smaller the information entropy of the index, the smaller the chaos degree in the system, the higher the order degree in the system, the more effective information provided by the index, the greater the contribution of the index to the system, the more significant effect in the comprehensive index evaluation, and the higher the entropy weight of the index. The calculation process of entropy weighting method is as follows.

①Calculate the $j$ indicator proportion of $i$ federal subject ($p_{ij}$):

$$p_{ij} = x_{ij} \bigg/ \sum_{i=1}^{m} x_{ij}; \tag{2}$$

②Calculate the entropy of the $j$ indicator ($e_j$):

$$e_j = (-1/\ln m) \sum_{i=1}^{m} p_{ij} \times \ln p_{ij}; \tag{3}$$

③Calculate the entropy weight of the $j$ indicator ($w_j$):

$$w_j = (1 - e_j) \bigg/ \sum_{j=1}^{n} (1 - e_j); \tag{4}$$

where $x_{ij}$ is the original $j$ index value of the $i$ federal subject. $m$ is the number of federal subjects. $n$ is the number of indicators.

*(2) Coefficient variation weighting method.* Coefficient variation is the proportion of the standard deviation to the average value, which considers the influence of the dispersion degree and average level of variable values. The greater the difference value of the same indicator in different federal subjects, the greater the weight of the indicator. This indicator could better reflect the relative disparity in different federal subjects, and it has greater impact on the whole urbanization system or eco-environment system. The calculation process of coefficient variation weighting method is as follows.

①Calculate the mean value of the $j$ indicator ($\bar{x}_j$) and the standard deviation of the $j$ indicator ($S_j$).

②Calculate the coefficient variation of the $j$ indicator ($CV_j$):

$$CV_j = S_j / \bar{x}_j; \tag{5}$$

③Calculate the coefficient variation weight of the $j$ indicator ($u_j$):

$$u_j = CV_j \Big/ \sum_{j=1}^{n} CV_j; \tag{6}$$

where $n$ is the number of indicators.

*(3) Comprehensive weighting method.* Two objective weights $w_j$ and $u_j$ of the indicator are calculated by entropy weighting method and coefficient variation weighting method respectively. The comprehensive weight value of the $j$ indicator ($p_j$) is calculated by the comprehensive weight method to reduce the errors of the two objective weight methods. The calculation process of comprehensive weighting method is as follows.

$$p_j = \sqrt{w_j \times u_j} \tag{7}$$

where $p_j$ is the comprehensive weight of $j$ indicator. $w_j$ is the entropy weight of $j$ indicator. $u_j$ is the coefficient variation weight of $j$ indicator.

**Development level model.** Based on the development level model of linear weighting method, we calculate the population urbanization level, economic urbanization level, social urbanization level, comprehensive urbanization level, eco-environment pressure level, eco-environment state level, eco-environment response level, comprehensive eco-environment level of 21 federal subjects in eastern Russia during 2005–2018. The calculation process of development level model is as follows.

$$W_i = \sum_{j=1}^{n}(w_j \times X_{ij}) \qquad U_i = \sum_{j=1}^{n}(u_j \times X_{ij}) \qquad P_i = \sqrt{W_i \times U_i} \tag{8}$$

where $W_i$ is the urbanization level or eco-environment level of the $i$ federal subject calculated by entropy weighting method. $U_i$ is the urbanization level or eco-environment level of the $i$ federal subject calculated by coefficient variation weighting method. $P_i$ is the comprehensive urbanization level or comprehensive eco-environment level of the $i$ federal subject.

**Coupling coordination model.** *(1) Coupling degree model.* There is a strong interaction between the urbanization development and the eco-environment development. Referring to the capacity coupling coefficient in the physics, a coupling model of urbanization and eco-environment suitable for eastern Russia is constructed. This model could characterize the "destruction" effect or "improvement" effect of the eco-environment in the process of urbanization development, and the "containment" effect or "promotion" effect of the urbanization in

the process of eco-environment development. Through the process of the stress, adaptation, interaction among the urbanization indicators and eco-environment indicators, the optimal sustainable development situation of urbanization system and eco-environment system will be achieved. The calculation process of coupling degree model is as follows.

$$C = \left[ \frac{P_1 \times P_2}{(P_1 + P_2)^2} \right]^{1/2} \tag{9}$$

where $P_1$ is the urbanization development level of the $i$ federal subject. $P_2$ is the eco-environment development level of the $i$ federal subject. $C$ is the coupling degree of the urbanization development level and eco-environment development level of the $i$ federal subject.

*(2) Coordinated development degree model.* On the basis of coupling degree model, in order to truly study the interaction intensity, overall coordination situation between urbanization development and eco-environment development in eastern Russia, a coordinated development degree model of urbanization and eco-environment suitable for eastern Russia is constructed. The calculation process of coordinated development degree model is as follows.

$$D = \sqrt{C \times T} \qquad T = \alpha P_1 + \beta P_2 \tag{10}$$

where $P_1$ is the urbanization development level of the $i$ federal subject. $P_2$ is the eco-environment development level of the $i$ federal subject. $D$ is the coordinated development degree of urbanization level and eco-environment level of $i$ federal subject, with the value of [0,1]. The closer the $D$ value is to 0, the stronger the stress effect between urbanization and co-environment, the weaker the interaction effect between urbanization and eco-environment. The closer the $D$ value is to 1, the stronger the coordinated effect between urbanization and eco-environment, the weaker the stress effect between urbanization and eco-environment. $T$ is the comprehensive coordination index of urbanization and eco-environment of $i$ federal subject. $\alpha$, $\beta$ respectively are the weight coefficients of urbanization and eco-environment. We believe that the urbanization and eco-environment are in the same important position, and $\alpha = \beta = 0.5$. According to the references [21–23], we classify the coordinated development types of urbanization and eco-environment into four categories and twelve subcategories (Table 3).

**Table 3. Classification of coordinated development stages of urbanization and eco-environment.**

| First-Level | Second-Level (Scope of $D$) | Third-Level(Comparison of $P_1$ and $P_2$) | Symbol |
|---|---|---|---|
| Coordinated development | High-level coordinated development ($0.8<D\leq1.0$) | Eco-environment backwardness ($P_1$-$P_2$>0.1) | I1 |
| | | Urbanization backwardness ($P_2$-$P_1$>0.1) | I2 |
| | | Urbanization and eco-environment balance ($0\leq|P_1$-$P_2|\leq0.1$) | I3 |
| | Basic coordinated development ($0.5<D\leq0.8$) | Eco-environment backwardness ($P_1$-$P_2$>0.1) | II1 |
| | | Urbanization backwardness ($P_2$-$P_1$>0.1) | II2 |
| | | Urbanization and eco-environment balance ($0\leq|P_1$-$P_2|\leq0.1$) | II3 |
| Uncoordinated development | Basic uncoordinated development ($0.3<D\leq0.5$) | Eco-environment backwardness ($P_1$-$P_2$>0.1) | III1 |
| | | Urbanization backwardness ($P_2$-$P_1$>0.1) | III2 |
| | | Urbanization and eco-environment balance ($0\leq|P_1$-$P_2|\leq0.1$) | III3 |
| | Serious uncoordinated development ($0<D\leq0.3$) | Eco-environment backwardness ($P_1$-$P_2$>0.1) | IV1 |
| | | Urbanization backwardness ($P_2$-$P_1$>0.1) | IV2 |
| | | Urbanization and eco-environment balance ($0\leq|P_1$-$P_2|\leq0.1$) | IV3 |

Note: Eco-environment backwardness means that the development level of eco-environment is lower than that of urbanization. Urbanization backwardness means that the development level of urbanization is lower than that of eco-environment.

### Data sources

The data of urbanization indicators comes from the 《*РЕГИОНЫ РОССИИ ОСНОВНЫЕ ХАРАКТЕРИСТИКИ СУБЪЕКТОВ РОССИЙСКОЙ ФЕДЕРАЦИИ СТАТИСТИЧЕСКИЙ СБОРНИК*》, which is published on the official website of the Russian Bureau of Statistics for the period 2006–2019. The data of eco-environment indicators comes from the 《*РЕГИОНЫ РОССИИ СОЦИАЛЬНО-ЭКОНОМИЧЕСКИЕ ПОКАЗАТЕЛИ СТАТИСТИЧЕСКИЙ СБОРНИК*》, which is published on the official website of the Russian Bureau of Statistics for the period 2006–2019.

## Results of urbanization and eco-environment

### Development level of urbanization and eco-environment

**Urbanization.** From 2005 to 2018, compared with the impacts of population urbanization (with a comprehensive weight of 0.34) and social urbanization (with a comprehensive weight of 0.22), the economic urbanization (with a comprehensive weight of 0.43) has the strongest impact on the urbanization system. From 2005 to 2018, the population urbanization rate of eastern Russia has only increased from 68.37% to 69.31%, which is far lower than that of the Russian average rate. The eastern Russia has serious population loss phenomenon and it also has relatively low quality labor force. However, the eastern Russia has a vast area and rich energy resources. Relying on the export of energy resources to obtain a large amount of foreign exchange income, the mining industry, metallurgy industry, electric power industry, automobile industry, machinery manufacturing industry, computer industry, atomic energy industry, consumer goods industry and synthetic material industry has become the main driving force for promoting the economic urbanization in eastern Russia.

The population urbanization development level, economic urbanization development level, social urbanization development level, and comprehensive urbanization development level are calculated based on the development level model of linear weighting method. From 2005 to 2018, the comprehensive urbanization level of eastern Russia has shown the increasing trend (from 4.38 to 7.28). The economic urbanization level of eastern Russia has also shown the increasing trend, which is consistent with the comprehensive urbanization trend. The population urbanization level of eastern Russia has shown a stable trend, and the social urbanization level of eastern Russia has shown a slight increasing trend (Fig 2). The population urbanization level, economic urbanization level, social urbanization level, and comprehensive urbanization level of the Siberian Federal District are all higher than those of the Far East Federal District (Fig 3). The federal subjects with high urbanization development levels are mainly distributed in the Novosibirsk Region, Krasnoyarsk Territory, Kemerovo Region, Altay Territory and Irkutsk Region. The federal subjects with low ones are mainly located in the Chukotka Autonomous Area, Jewish Autonomous Area, Magadan Region and Kamchatka Territory (Table 4).

In recent years, the urbanization level of eastern Russia has shown an increasing trend. In terms of population urbanization, the government has introduced a preferential assistance mechanism to stabilize the local population and it has also attracted foreign high-tech talents through the land donation, housing donation and special business methods, so as to vigorously increase the total population and non-agricultural labor force. In terms of economic urbanization, according to the different urban functional structures in different regions, the corresponding industrial sectors has been planned, such as agriculture, forestry, fishery, energy, manufacturing, water conservancy, metallurgy, chemical industry, mining, construction, transportation and tourism. For example, the government has built the wood deep-processing industrial clusters in the south areas of Far East Federal District, the fishery industrial clusters

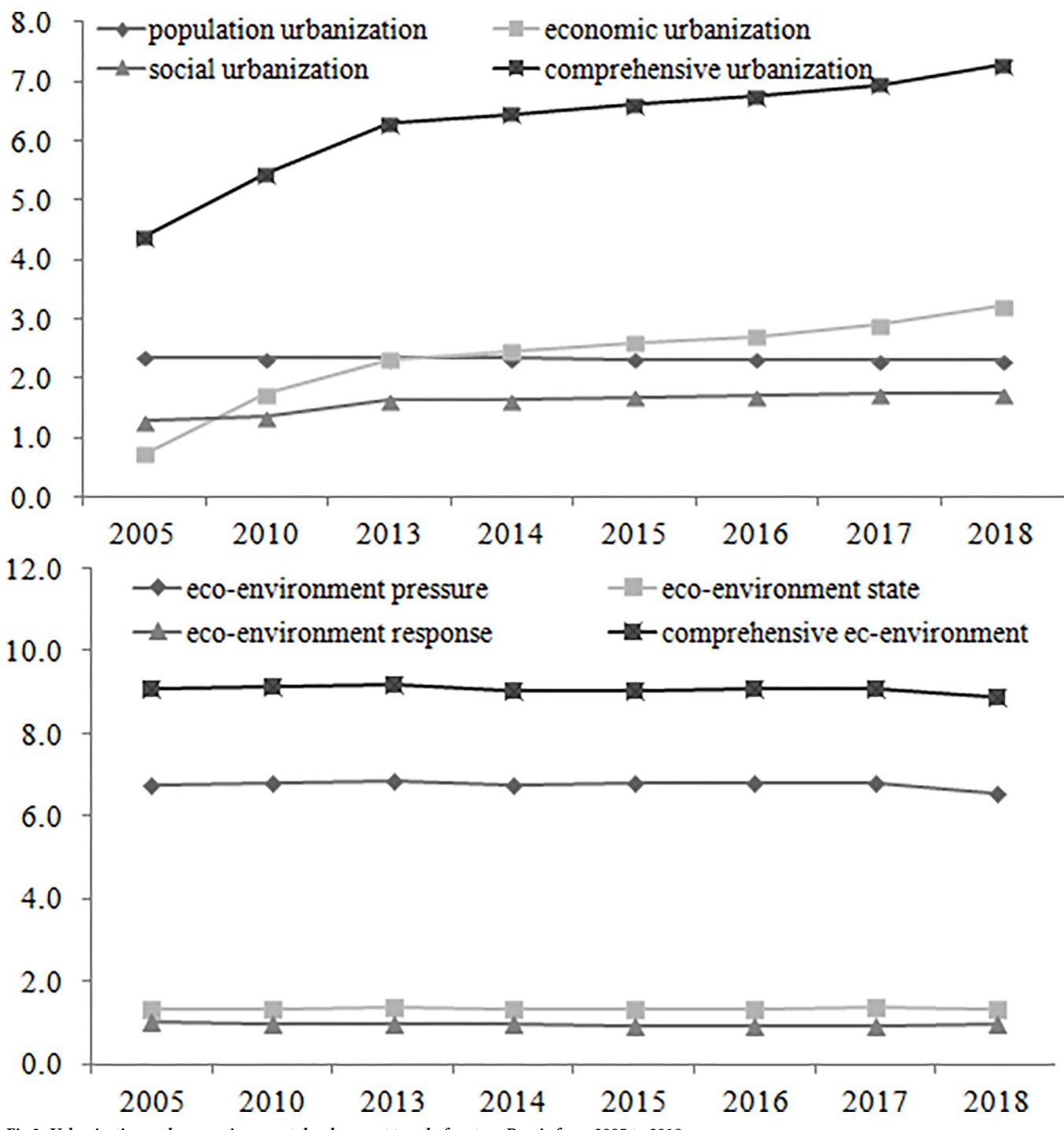

**Fig 2. Urbanization and eco-environment development trend of eastern Russia from 2005 to 2018.**

in the coastal areas of Far East Federal District, the tourism industrial clusters in Baikal Lake, Amur River and Primorsky Territory, so as to promote various economic factors aggregation in the non-agricultural industries. In terms of social urbanization, the basic infrastructures have been planned, such as the transportation, energy and telecommunication infrastructures. For example, the Siberian Railway will be upgraded, the Far East Federal District's roads will

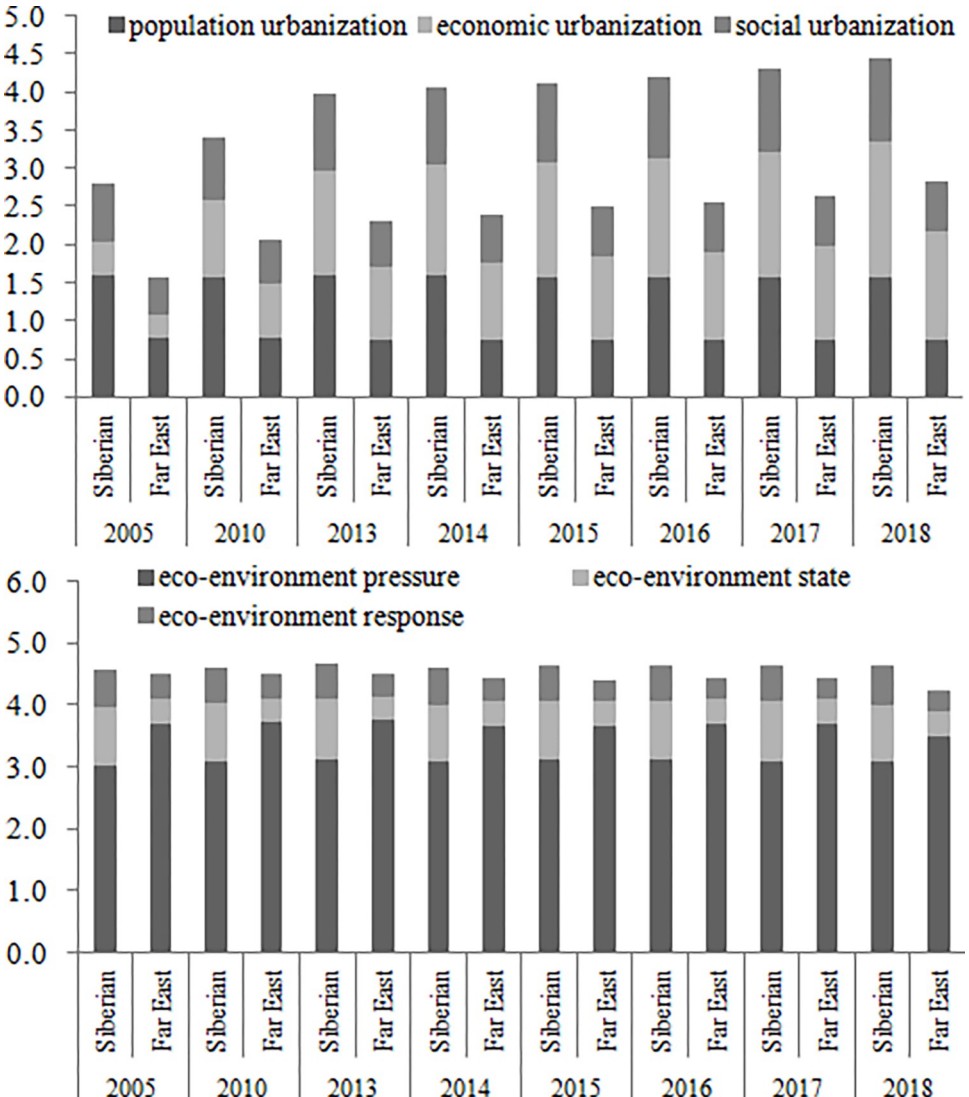

**Fig 3. Urbanization and eco-environment development trend of Siberian Federal District and Far East Federal District from 2005 to 2018.**

be integrated into the Russian road network system, the international aviation hubs such as Khabarovsk and Vladivostok will be built, the oil and gas pipeline system between eastern Siberian Federal District and the Pacific Ocean will be constructd, the high-speed communication lines among the Primorsky Territory, China and Japan will be built.

**Eco-environment.** From 2005 to 2018, compared with the impact of eco-environmental response (with a comprehensive weight of 0.26), the eco-environment pressure (with a comprehensive weight of 0.38) and eco-environment state (with a comprehensive weight of 0.36) have stronger impacts on the eco-environment system. In the process of urbanization development, the eco-environment protection needs to be further strengthened in eastern Russia. Eastern Russia has a large number of traditional industrial cities. In the process of exploration, mining and transportation, various minerals, oil and gas resources cause damage to the vegetation, surface water and groundwater. Besides, solid waste which comes from the industrial

**Table 4. Urbanization, eco-environment and their coupling coordination of each federal subject in eastern Russia in 2018.**

| Federal subjects | Urbanization | | | | Eco-environment | | | | Coupling degree | Coordinated development degree | Coupling coordination stage |
|---|---|---|---|---|---|---|---|---|---|---|---|
| | Population | Economic | Social | Comprehensive | Pressure | State | Response | Comprehensive | | | |
| Republic of Altay | 0.02 | 0.02 | 0.03 | 0.07 | 0.36 | 0.01 | 0.00 | 0.37 | 0.37 | 0.28 | Serious uncoordinated |
| Republic of Tyva | 0.03 | 0.02 | 0.04 | 0.09 | 0.35 | 0.01 | 0.01 | 0.38 | 0.39 | 0.30 | Basic uncoordinated |
| Republic of Khakasia | 0.08 | 0.05 | 0.05 | 0.18 | 0.35 | 0.02 | 0.02 | 0.38 | 0.47 | 0.36 | Basic uncoordinated |
| Altay Territory | 0.21 | 0.17 | 0.18 | 0.55 | 0.35 | 0.22 | 0.03 | 0.60 | 0.50 | 0.54 | Basic coordinated |
| Krasnoyarsk Territory | 0.20 | 0.42 | 0.17 | 0.79 | 0.13 | 0.19 | 0.16 | 0.48 | 0.49 | 0.55 | Basic coordinated |
| Irkutsk Region | 0.17 | 0.29 | 0.16 | 0.62 | 0.26 | 0.10 | 0.15 | 0.50 | 0.50 | 0.53 | Basic coordinated |
| Kemerovo Region | 0.31 | 0.27 | 0.14 | 0.72 | 0.26 | 0.09 | 0.11 | 0.47 | 0.49 | 0.54 | Basic coordinated |
| Novosibirsk Region | 0.26 | 0.28 | 0.15 | 0.69 | 0.34 | 0.12 | 0.03 | 0.49 | 0.49 | 0.54 | Basic coordinated |
| Omsk Region | 0.19 | 0.17 | 0.11 | 0.47 | 0.33 | 0.14 | 0.06 | 0.53 | 0.50 | 0.50 | Basic coordinated |
| Tomsk Region | 0.09 | 0.11 | 0.07 | 0.27 | 0.34 | 0.04 | 0.03 | 0.42 | 0.49 | 0.41 | Basic uncoordinated |
| Republic of Buryatia | 0.07 | 0.19 | 0.07 | 0.33 | 0.35 | 0.04 | 0.04 | 0.42 | 0.50 | 0.43 | Basic uncoordinated |
| Republic of Sakha (Yakutia) | 0.07 | 0.29 | 0.08 | 0.44 | 0.18 | 0.09 | 0.04 | 0.30 | 0.49 | 0.43 | Basic uncoordinated |
| Zabaikalsk Territory | 0.08 | 0.10 | 0.09 | 0.27 | 0.33 | 0.04 | 0.04 | 0.40 | 0.49 | 0.40 | Basic uncoordinated |
| Kamchatka Territory | 0.03 | 0.06 | 0.03 | 0.13 | 0.35 | 0.02 | 0.01 | 0.38 | 0.43 | 0.33 | Basic uncoordinated |
| Primorsky Territory | 0.19 | 0.21 | 0.11 | 0.51 | 0.31 | 0.05 | 0.08 | 0.44 | 0.50 | 0.49 | Basic uncoordinated |
| Khabarovsk Territory | 0.10 | 0.18 | 0.08 | 0.36 | 0.32 | 0.05 | 0.08 | 0.45 | 0.50 | 0.45 | Basic uncoordinated |
| Amur Region | 0.07 | 0.14 | 0.07 | 0.27 | 0.25 | 0.06 | 0.04 | 0.35 | 0.50 | 0.39 | Basic uncoordinated |
| Magadan Region | 0.02 | 0.04 | 0.03 | 0.10 | 0.35 | 0.02 | 0.01 | 0.39 | 0.40 | 0.31 | Basic uncoordinated |
| Sakhalin Region | 0.07 | 0.19 | 0.04 | 0.30 | 0.35 | 0.02 | 0.01 | 0.38 | 0.50 | 0.41 | Basic uncoordinated |
| Jewish Autonomous Area | 0.04 | 0.02 | 0.03 | 0.08 | 0.35 | 0.01 | 0.01 | 0.37 | 0.38 | 0.29 | Serious uncoordinated |
| Chukotka Autonomous Area | 0.01 | 0.02 | 0.02 | 0.05 | 0.35 | 0.01 | 0.01 | 0.37 | 0.33 | 0.26 | Serious uncoordinated |

production pollutes air, water and land. These threaten the surrounding ecological environment, endanger the living environment, and reduce the environmental carrying capacity.

The eco-environment pressure level, eco-environment state level, eco-environmental response level, and comprehensive urbanization development level are calculated based on the development level model of linear weighting method. From 2005 to 2018, the comprehensive

eco-environment level of eastern Russia has shown the slight decreasing trend. The eco-environment pressure level, eco-environment state level, and eco-environmental response level have also shown the decreasing trend (Fig 2). The eco-environment pressure level, eco-environment state level, eco-environmental response level, and comprehensive eco-environment level of the Siberian Federal District are all balanced to those of the Far East Federal District (Fig 3). The federal subjects with high eco-environment development levels are mainly distributed in the Altay Territory, Omsk Region, Novosibirsk Region, Irkutsk Region, Krasnoyarsk Territory and Kemerovo Region. The federal subjects with low ones are mainly located in the Republic of Sakha(Yakutia), Amur Region, Chukotka Autonomous Area, Jewish Autonomous Area and Sakhalin Region (Table 4).

The heavy industry development caused a series of environmental problems for eastern Russia in the Soviet Union period. In recent years, in order to promote the urbanization development and market economy, the contradiction has become increasingly serious between natural resource development and ecological environment protection in eastern Russia. The production activities of industrial enterprises (fuel power industry, electric power industry, metallurgical industry, thermal power stations, and motor vehicles) emit a large amount of carbon dioxide, sulfur oxide, phenol, formaldehyde, ammonia, lead, suspended particles and other pollutants into the air. The continuous exploitation of oil, gas, coal and other resources has led to a long-term high-level of carbon density in the air. Agricultural and industrial production, mining development and public utilities construction all discharge a large amount of pollutants into the rivers, which causes surface water pollution and groundwater pollution along Yenisei River, Amur River, Lena River, Obi river, Ertis River, Tobor River, Tula River and Tommy River. In addition, Krasnoyarsk Territory, Republic of Buryatia, Primorsky Territory, Sakhalin Region, and Kemerovo Region have high-risk land pollution of heavy metal salts. Kamchatka Territory has missile fuel with extremely aggressive toxic substances. Primorsky Territory has decommissioned nuclear submarines remained to be destroyed. Some areas of Magadan Region have large amount of dangerous element Radon. Under the action of air, these diffuse radioactive nuclear wastes cause serious pollution to the eco-environment in eastern Russia.

## Temporal and spatial pattern of urbanization and eco-environment

Based on the 3D perspective, the "global trend analysis" in ArcGIS was used to study the urbanization development level changes and eco-environment development level changes in eastern Russia (Fig 4). The X-axis shows the east-west direction of the whole territory of eastern Russia (the arrow points to the East), the Y-axis shows the north-south direction of the whole territory of eastern Russia (the arrow points to the North), and the height of each vertical line of the Z-axis shows the urbanization development level or eco-environment development level of each federal subject. From 2005 to 2018, the urbanization level and eco-environment level both show the spatial characteristics of "High West, Low East" in the east-west direction, which is reflected in the decreasing trend of the first-order function from west to east. From 2005 to 2018, the difference of urbanization level shows an expanding trend in the north-south direction, which is reflected in the steady decreasing trend of the first-order function changing into a steep decreasing trend of the first-order function. The difference of eco-environment level shows a decreasing trend in the north-south direction, which is reflected in the steep decreasing trend of the first-order function changing into the inverse U-shaped trend of second-order function.

Based on the 2D perspective, the "spatial analysis function" in ArcGIS was used to study the urbanization development level changes and eco-environment development level changes in

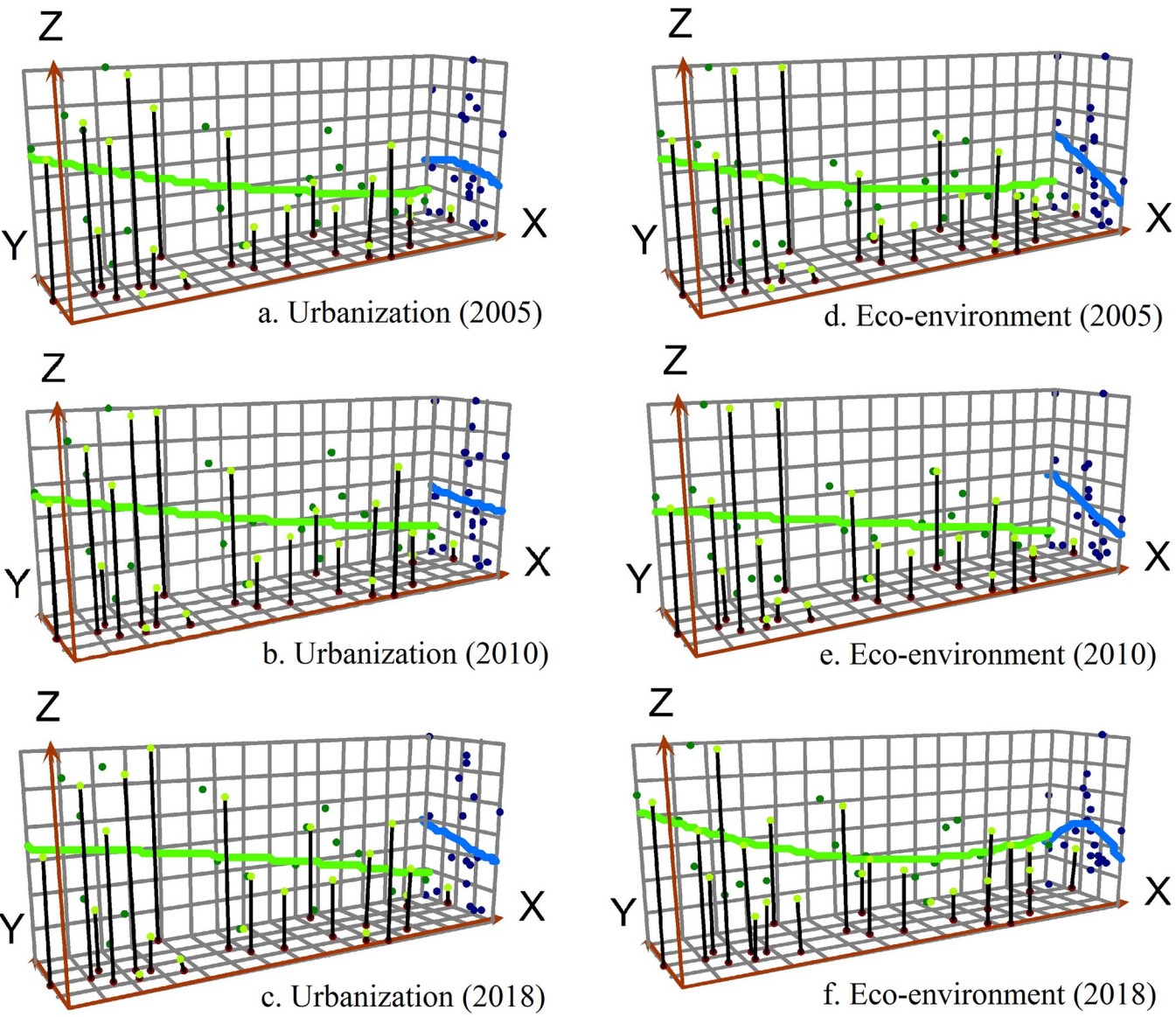

**Fig 4. Global trend changes of urbanization and eco-environment in eastern Russia from 2005 to 2018.**

eastern Russia (Fig 5). From 2005 to 2018, the urbanization level of the eastern Russian subjects shows the increasing trend, and the ecological environment level of the eastern Russian federal subjects shows the decreasing trend, which both characterize the "High West, Low East" pattern. As Irkutsk Region the boundary, the urbanization level and the eco-environment level of the Siberian Federal District are obviously stronger than those of the Far East Federal District. The areas with high urbanization level and high eco-environment level are mainly distributed in the series areas of Novosibirsk Region- Altay Territory- Kemerovo Region- Krasnoyarsk Territory- Irkutsk Region. The areas with low ones are mainly located in the series areas of Republic of Altay- Republic of Tyva in the south of Siberian Federal District, and Kamchatka Territory- Chukotka Autonomous Area- Magadan Region in the northeast of Far East Federal District.

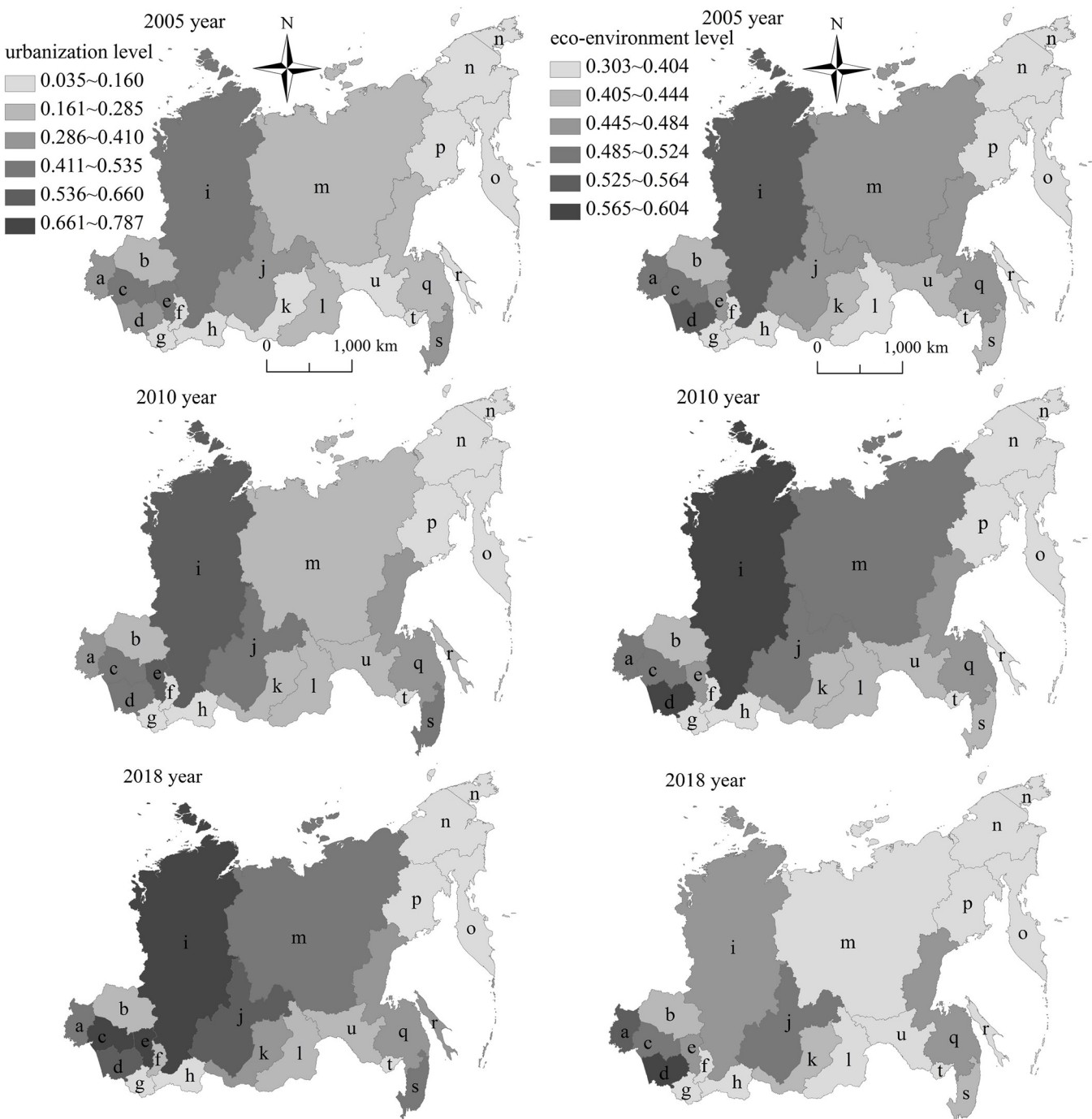

**Fig 5. Spatial pattern changes of urbanization and eco-environment in eastern Russia from 2005 to 2018.** Note: a Omsk Region. b Tomsk Region. c Novosibirsk Region. d Altay Territory. e Kemerovo Region. f Republic of Khakasia. g Republic of Altay. h Republic of Tyva. i Krasnoyarsk Territory. j Irkutsk Region. k Republic of Buryatia. l Zabaikalsk Territory. m Republic of Sakha(Yakutia). n Chukotka Autonomous Area. o Kamchatka Territory. p Magadan Region. q Khabarovsk Territory. r Sakhalin Region. s Primorsky Territory. t Jewish Autonomous Area. u Amur Region. This drawing has not been previously copyrighted. The authors created the image themselves.

The urbanization development level and eco-environment development level of the Siberian Federal District are stronger than those of the Far East Federal District. Historically, Siberian Federal District was the economic and industrial center in Russia, which had dense

population, strong attraction immigrants, high quality labor, strong economic strength, diversified industrial structure, outstanding foreign trade, strong scientific and technological innovation and dense transportation network. However, the development of the Far East Federal District is mainly for the territorial security and military defense. In 1916, the operating of the Siberian Railway brought real opportunities to the development of Far East Federal District. But after the disintegration of the Soviet Union, the unbalanced natural resources, lost human resources, underdeveloped economy, shrinking military industry, weak foreign cooperation, outdated technology and equipment, backward high-tech level, deep resource curse effect, imperfect social infrastructure all had bad effects on the development of Far East Federal District [30–32]. In recent years, the eco-environment problems has become more and more severe because of the oil and natural gas over-exploitation, modern pollution prevention facilities lacking, forest coverage rate decreasing, water resources shortage, air and land pollution in the Far East Federal District. In the future, under the shift of the world economic center to the Asia Pacific region, the Far East Federal District has planned the southern urban agglomeration and the island economic belt with the guidance of policies such as the 《Federal Special Outline for Economic and Social Development of the Far East Federal District and Zabaikalsk Territory before 2013》 and 《Economic and Social Development Strategy of the Far East Federal District and Baikal Region before 2025》. It has also planned to build many new infrastructure in the fields of transportation, communications, energy resources, environmental protection, which continuously consolidates its economic status in the Asia Pacific region. It will become the important reserve economic security district of the Siberian Federal District. The difference of the urbanization level and eco-environment level may be decreased between the Siberian Federal District and Far East Federal District.

## Results of coupling coordination between urbanization and eco-environment

### Types of coupling coordination degree

The coupling coordinated development degree of the urbanization and eco-environment levels was measured based on the coupling coordination model in eastern Russia. From 2005 to 2018, the coupling coordination development degree of urbanization and eco-environment has shown the increasing trend, with $C$ value rising from 0.43 to 0.46, $T$ value rising from 0.32 to 0.39, and $D$ value rising from 0.37 to 0.42 in eastern Russia. However, the urbanization and eco-environment are in the basic uncoordinated development stage, which change from the urbanization backwardness to the urbanization and eco-environment balance in its internal system. The urbanization and ecological environment of the Siberian Federal District and Far East Federal District are also in the basic uncoordinated development stage, with $C$ value: Siberian Federal District (0.47) > Far East Federal District (0.46), $T$ value Siberian Federal District (0.45) > Far East Federal District (0.32), $D$ value: Siberian Federal District (0.46) > Far East Federal District t (0.38). The coupling coordination development degree of Siberian Federal District is slightly stronger than that of the Far East Federal District. The Siberian Federal District changes from the urbanization backwardness to the urbanization and eco-environment balance in its internal system, the Far East Federal District is still the urbanization backwardness in its internal system. In 2018, the proportion of federal subjects with high-level coordination development, basic coordination development, basic uncoordinated development, serious uncoordinated development is 0.0%, 28.6%, 57.1%, 14.3% respectively. Most federal subjects are in the basic uncoordinated development stage and least federal subjects belong to the high-level coordination development stage (Tables 4 and 5). Krasnoyarsk Territory, Novosibirsk Region, Kemerovo Region, Altay Territory and Irkutsk Region have high coupling

**Table 5. Coupling coordination stage of urbanization and eco-environment in each federal subject in eastern Russia.**

| Federal subjects | 2005 | 2010 | 2013 | 2014 | 2015 | 2016 | 2017 | 2018 |
|---|---|---|---|---|---|---|---|---|
| Republic of Altay | IV2 | IV2 | IV2 | IV2 | IV2 | IV2 | IV2 | IV2 |
| Republic of Tyva | IV2 | IV2 | IV2 | IV2 | IV2 | IV2 | IV2 | III2 |
| Republic of Khakasia | III2 | III2 | III2 | III2 | III2 | III2 | III2 | III2 |
| Altay Territory | III2 | III2 | II3 | II3 | II3 | II3 | II3 | II3 |
| Krasnoyarsk Territory | III2 | II3 | II1 | II1 | II1 | II1 | II1 | II1 |
| Irkutsk Region | III3 | III3 | II3 | II3 | II3 | II1 | II1 | II1 |
| Kemerovo Region | III3 | II1 | II1 | II1 | II1 | II1 | II1 | II1 |
| Novosibirsk Region | III3 | II3 | II1 | II1 | II1 | II1 | II1 | II1 |
| Omsk Region | III2 | III2 | III3 | III3 | III3 | III3 | III3 | II3 |
| Tomsk Region | III2 | III2 | III2 | III2 | III2 | III2 | III2 | III2 |
| Republic of Buryatia | III2 | III2 | III2 | III2 | III2 | III2 | III2 | III3 |
| Republic of Sakha (Yakutia) | III3 | III2 | III2 | III2 | III2 | III3 | III3 | III1 |
| Zabaikalsk Territory | III2 | III2 | III2 | III2 | III2 | III2 | III2 | III2 |
| Kamchatka Territory | IV2 | III2 | III2 | III2 | III2 | III2 | III2 | III2 |
| Primorsky Territory | III2 | III3 | III3 | III3 | III3 | III3 | III3 | III3 |
| Khabarovsk Territory | III2 | III2 | III2 | III2 | III2 | III2 | III2 | III3 |
| Amur Region | III2 | III2 | III2 | III2 | III2 | III2 | III2 | III3 |
| Magadan Region | IV2 | IV2 | IV2 | IV2 | III2 | III2 | III2 | III2 |
| Sakhalin Region | III2 | III2 | III2 | III3 | III3 | III3 | III3 | III3 |
| Jewish Autonomous Area | IV2 | IV2 | IV2 | IV2 | IV2 | IV2 | IV2 | IV2 |
| Chukotka Autonomous Area | IV2 | IV2 | IV2 | IV2 | IV2 | IV2 | IV2 | IV2 |

coordination development degree (*D*> 0.50), with the potential to achieve the high-level coordination development stage in the future. Other federal subjects are still in the uncoordinated development stage, with *D* value less than 0.50.

According to Table 5, in 2005, eighteen federal subjects belonged to the urbanization backwardness type, and three federal subjects belonged to the urbanization and eco-environment balance type in eastern Russia. Urbanization backwardness was the main type, and urbanization had a restrictive effect on the eco-environment. In 2010, sixteen federal subjects belonged to the urbanization backwardness type, four federal subjects belonged to the urbanization and eco-environment balance type, and only one federal subject belonged to the eco-environment backwardness type in eastern Russia. Urbanization backwardness was still the main type. In 2018, nine federal subjects belonged to the urbanization backwardness type, seven federal subjects belonged to the urbanization and eco-environment balance type, and five federal subjects belonged to the eco-environment backwardness type in eastern Russia. The stress effect of urbanization has gradually decreasing on the eco-environment. Although the urbanization development has brought some harm to the eco-environment, Russia has always issued some laws, such as the 《Natural Environment Protection Law of the Russian Federation》, 《the National Ecological Environment Law of the Russian Federation》, 《the Law of the Special Nature Protection Zone of the Russian Federation》. It has also signed the 《Convention on International Trade in Endangered Species of Wild Animal and Plant Species》, 《Convention on Biological Diversity》, 《Convention on Wetlands》, 《the United Nations Convention on the Law of the Sea》 to work on the restoration and governance of the eco-environment. Eastern Russia has also implemented many eco-environment security measures, such as improving the people's awareness of eco-environment protection, improving laws and regulations on eco-environment protection, strengthening the functions of corresponding administrative organs, establishing economic laws and regulations on environmental

protection, implementing ecological insurance mechanism, and establishing special nature protection zones, so as to avoid the overload carrying capacity problems of the rapid urbanization development bringing to the resources and environment.

## Temporal and spatial pattern of coupling coordination degree

From the 3D perspective, the coordinated development degree of urbanization and eco-environment shows the unbalanced pattern characteristics of "High West, Low East" and "High Center, Low South, Low North" in eastern Russia. The difference of coordinated development degree of urbanization and eco-environment in the east-west direction is greater than that in the north-south direction (Fig 6). From the 2D perspective, the coordinated development degree of urbanization and eco-environment shows an increasing trend, which is similar to the pattern of urbanization level and eco-environment level, presenting the unbalanced differentiation characteristics of "High West, Low East" in eastern Russia. The Siberian Federal District has stronger coordinated development degree of urbanization and eco-environment compared with the Far East Federal District. The federal subjects with high coordinated development degree of urbanization and eco-environment are mainly distributed in the serious areas of Omsk Region- Novosibirsk Region—Altay Territory- Kemerovo Region- Krasnoyarsk Territory- Irkutsk Region. The federal subjects with low ones are located in the Republic of Altay in the south of Siberian Federal District, and Chukotka Autonomous Area in the northeast of Far East Federal District (Fig 7).

From 2005 to 2018, Novosibirsk Region, Altay Territory, Kemerovo Region, Krasnoyarsk Territory and Irkutsk Region change from the basic uncoordinated development type into the basic coordinated development type, which their internal system changes from the urbanization and eco-environment balance into the eco-environment backwardness. As the economic development poles of whole eastern Russia, these federal subjects have high urbanization level and high eco-environment level in eastern Russia. Relying on good regional advantages, rich resources and energy, diversified economic structure, dense transportation network, perfect logistics facilities, comprehensive security system, these federal subjects undertake the economic factors transfer from western Europe, becoming the gathering places of heavy industries such as machine manufacturing, energy processing, coal mining and the economic zone along Siberian railway corridor. While promoting the urbanization development levels, these cause a certain pressure on the eco-environment development. From 2005 to 2018, Primorsky Territory, Khabarovsk Territory, Republic of Buryatia, Amur Region, Sakhalin Region, Republic of Sakha(Yakutia) and Omsk Region are still in the basic uncoordinated development type, which their internal system changes from the urbanization backwardness to the urbanization and eco-environment balance. Primorsky Territory, Khabarovsk Territory, and Amur Region are the important economic growth areas of Far East Federal District, which have the important function of Russia's integration into the Asia Pacific region to promote the economic cooperation within the Sino-Russian border areas. Republic of Sakha(Yakutia) and Sakhalin Region have great exploitation potential in energy resources, which are not only the value subject supporting the development, but also the source power of economic urbanization development of some resource-based cities in the Far East Federal District. Omsk Region is the connecting district of western Europe and eastern Asia in Russia, and Republic of Buryatia is the border of China, Mongolia and Russia. Relying on location advantages, Omsk Region and Republic of Buryati have deepened the internal and external integration by creating superior internal and external markets. From 2005 to 2018, Zabaikalsk Territory, Tomsk Region and Republic of Khakasia are still in the basic uncoordinated development type, which their internal system is still in the urbanization backwardness type. Attracted by the neighboring federal

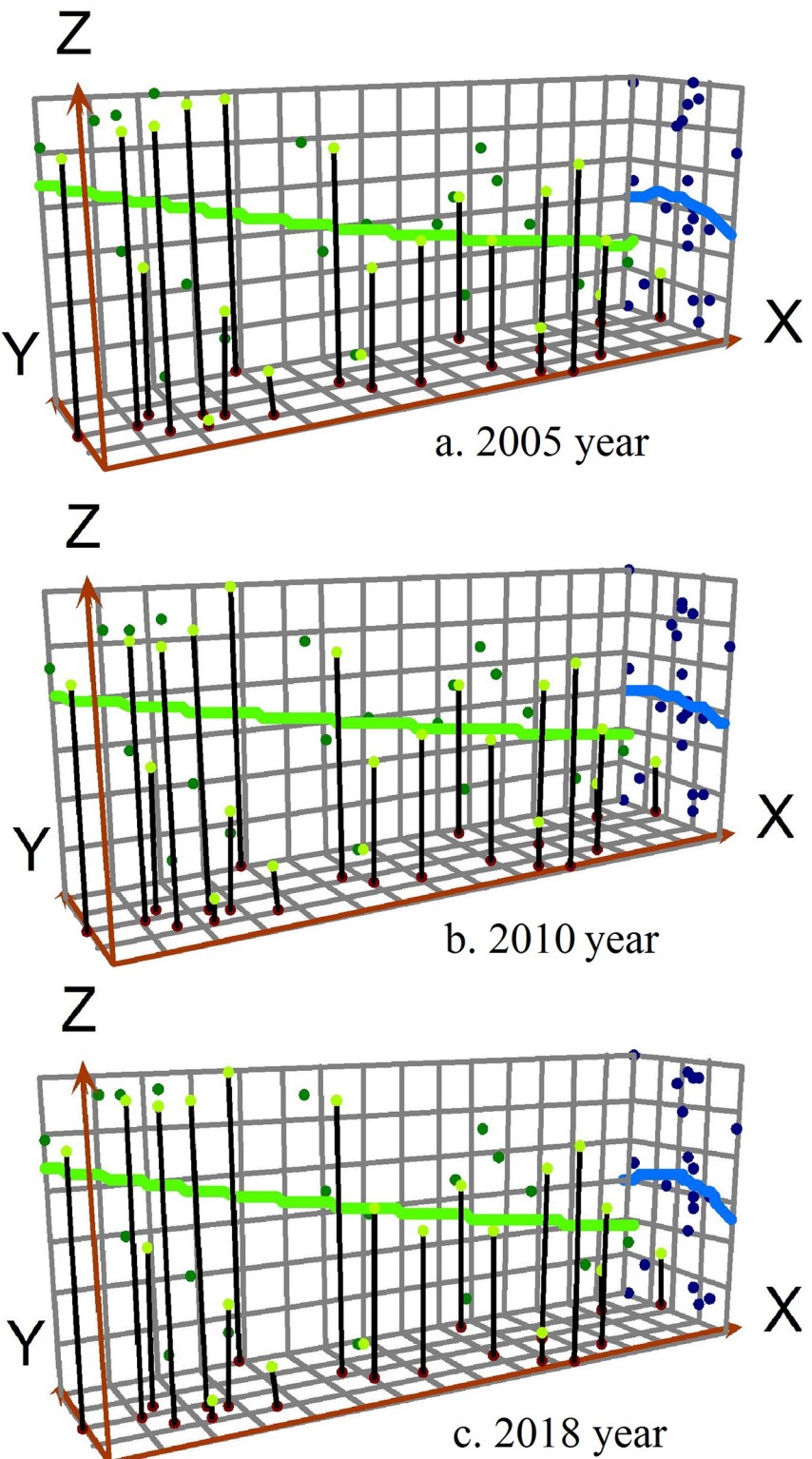

**Fig 6. Global trend changes of coupling coordination degree between urbanization and eco-environment in eastern Russia from 2005 to 2018.**

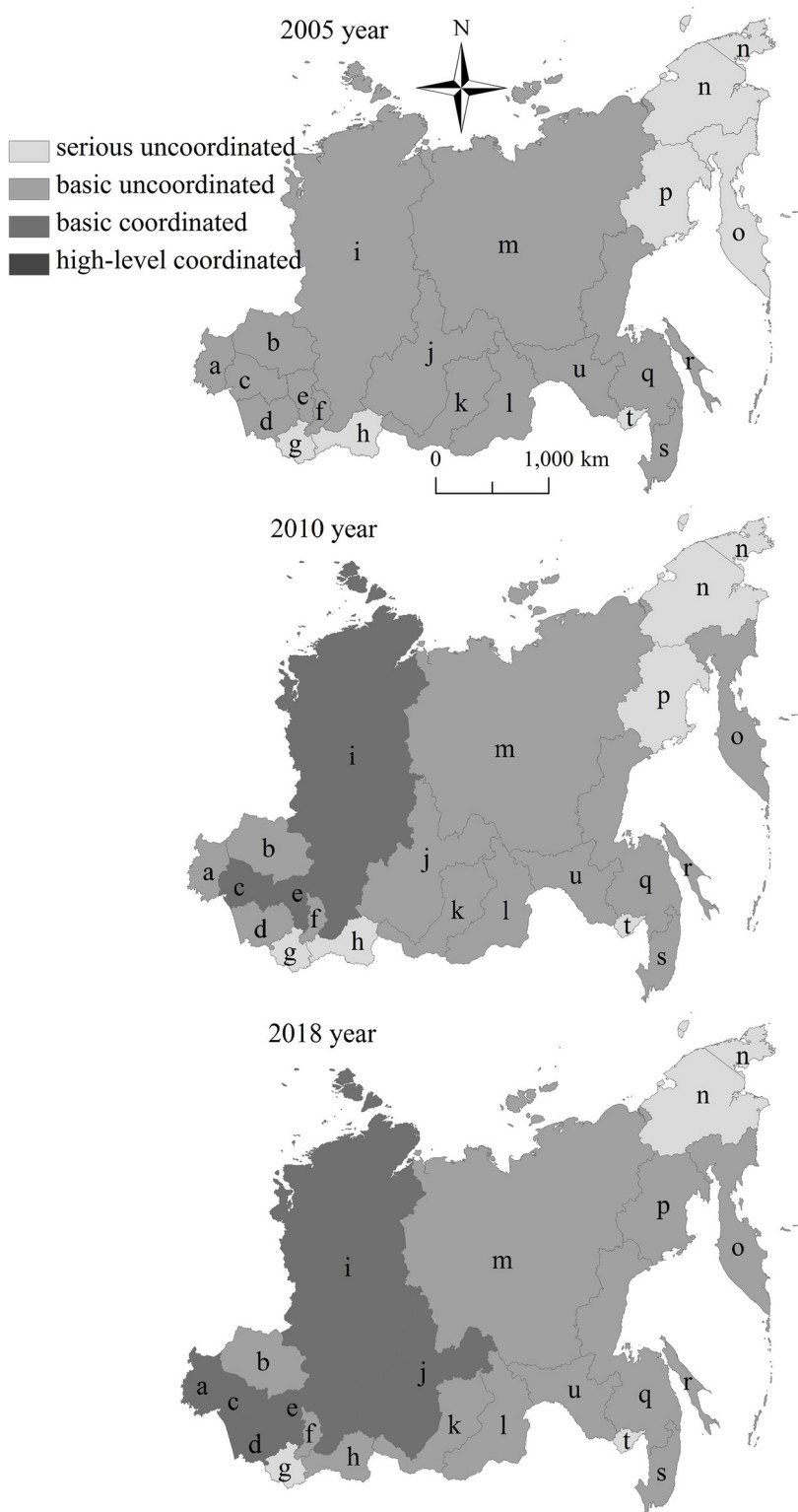

**Fig 7. Spatial pattern changes of coupling coordination degree between urbanization and eco-environment in eastern Russia from 2005 to 2018.** Note: a Omsk Region. b Tomsk Region. c Novosibirsk Region. d Altay Territory. e Kemerovo Region. f Republic of Khakasia. g Republic of Altay. h Republic of Tyva. i Krasnoyarsk Territory. j Irkutsk Region. k Republic of Buryatia. l Zabaikalsk Territory. m Republic of Sakha(Yakutia). n Chukotka Autonomous Area. o Kamchatka Territory. p Magadan Region. q Khabarovsk Territory. r Sakhalin Region. s Primorsky Territory. t Jewish

Autonomous Area. u Amur Region. This drawing has not been previously copyrighted. The authors created the image themselves.

subjects with superior economy, these federal subjects have weak agricultural and light industrial production, small energy industry scale, insufficient scientific and technological innovation ability, weak economic diversified development, excessive external market dependence, outflow of capital and shortage of funds. Therefore, the urbanization development of these federal subjects lags behind the eco-environment. From 2005 to 2018, Republic of Tyva, Kamchatka Territory and Magadan Region change from the serious uncoordinated development type into the basic uncoordinated development type, which their internal system is still in the urbanization backwardness type. Republic of Altay, Jewish Autonomous Area and Chukotka Autonomous Area are still in the serious uncoordinated development type, which their internal system is still in the urbanization backwardness type. These federal subjects are located in the marginal areas of eastern Russia. Limited by the severe climate conditions, low degree development, serious population outflow and poor transportation infrastructure, the resource and energy advantages and economic development potential have not been fully utilized. Although the urbanization level is low, the ecological environment is maintained well.

## Discussion

Compared with earlier studies [3–16], first we studied the coordinated development of urbanization and ecological environment in eastern Russia, which could make up for the deficiency of a single study that only studied urbanization or ecological environment before. Second, we combined with the Population- Economic- Sociology and Pressure- State- Response models to construct urbanization evaluation system and eco-environment evaluation system, which could have more scientificity, logicality, focalization compared with previous urbanization or eco-environment studies in eastern Russia. Third, in the process of index weight determination, the combination of entropy weighting and coefficient variation weighting could not only avoid the uncertainty of the subjective weighting method, but also could get rid of the defects of the single objective weighting method. Finally, as for the previous historical qualitative research, the quantitative and visual research need to be continuously tracked and updated on the coordinated development of urbanization and ecological environment in eastern Russia.

In recent years, the urbanization and eco-environment are in the basic uncoordinated development stage in eastern Russia. In the future, on the basis of solving the contradiction between resource development and eco-environment protection, eastern Russia should promote the urbanization quality improvement. It should also coordinate the economic and social development strategies, so as to promote the economic balance development and the eco-environment protection among twenty-one federal subjects. On the basis of increasing regional financial support, optimizing regional infrastructure, protecting regional environment, developing innovative economic model, optimizing industrial structure, and encouraging governments functions, it should create conditions for economic, social and eco-environment development, driving the development of marginal areas with advantageous areas to realize diversified development.

Siberian Federal District should build a modern economic metropolitan area with Novosibirsk Region as the core, radiating Altay Territory, Kemerovo Region, Krasnoyarsk Territory and Irkutsk Region. It could integrate production factors, implement innovation driven development strategy, promote economic transformation in resource-based areas, and vigorously develop modern service industry. At the same time, it should develop the Arctic Development Zone on the basis of strengthening resource and energy exploration, restoring Arctic routes

development, and protecting the original eco-environment. What's more, it should develop the Northern Development Belt, focusing on automobile manufacturing, chemical industry, electric power and other industries. It should promote the Southern Development Belt, innovatively transforming its traditional economic sectors so as to develop industrial and economic cooperation zones with Asia Pacific and Central Asia.

Far East Federal District should stabilize the local population, attracting foreign labor force on the basis of the common development of eco-environment protection and economic interests. It should focus on planning infrastructure such as energy, transportation and information, especially paying attention to the cooperation with Northeast Asia. In the northeast marginal areas Far East Federal District, it should get rid of the inherent anti-immigration and ethnic discrimination, improving the stable resettlement system for labor immigrants. It also should formulate economic and urban construction investment plans, restoring its economic activity in order to attract immigrants from other regions. In the south of Far East Federal District, it should develop the agricultural industrial complexes and tourism leisure clusters in Khabarovsk Territory and Primorsky Territory. It should use eco-environment protection raw materials to transform the industrial structure, developing its diversified advantages of innovative resources. It also should take advantage of the ports' advantages to enhance investment attraction, expanding production factors to other federal subject. In addition, it should deepen ecological and economic cooperation with northeast China, North Korea, South Korea and other neighboring countries, accelerating the planning and construction of transportation facilities synchronized with economic development.

## Conclusions

From 2005 to 2018, the urbanization development level has shown the increasing trend, the eco-environment development level has shown a slight decreasing trend, and their coupling coordination development degree has shown an increasing trend in eastern Russia. The urbanization and eco-environment are in the basic uncoordinated development stage, which change from the urbanization backwardness to the urbanization and eco-environment balance in its internal system. The urbanization levels with population, economic and social aspects of the Siberian Federal District are all higher than those of the Far East Federal District. The eco-environment levels with pressure, state and response aspects of the Siberian Federal District are all balanced to those of the Far East Federal District. The urbanization and ecological environment of the Siberian Federal District and Far East Federal District are in the basic uncoordinated development stage. Among them, Krasnoyarsk Territory, Novosibirsk Region, Kemerovo Region, Altay Territory and Irkutsk Region have high coupling coordination development degree, other federal subjects are still in the uncoordinated development stage.

Spatially, from the 3D perspective, the urbanization development level, the eco-environment development level and their coupling coordination development degree all show the spatial characteristics of "High West, Low East" and "High Center, Low South, Low North" in eastern Russia. From the 2D perspective, the urbanization development level, the eco-environment development level and their coupling coordination development degree characterize the "High West, Low East" pattern in eastern Russia. The federal subjects with high urbanization level, eco-environment level and their coupling coordination development degree are mainly distributed in the serious areas of Novosibirsk Region—Altay Territory- Kemerovo Region- Krasnoyarsk Territory- Irkutsk Region. The federal subjects with low ones are mainly located in the Republic of Altay in the south of Siberian Federal District, and Chukotka Autonomous Area in the northeast of Far East Federal District.

From 2005 to 2018, Novosibirsk Region, Altay Territory, Kemerovo Region, Krasnoyarsk Territory and Irkutsk Region change from the basic uncoordinated development type into the

basic coordinated development type, which their internal system changes from the urbanization and eco-environment balance into the eco-environment backwardness. Primorsky Territory, Khabarovsk Territory, Republic of Buryatia, Amur Region, Sakhalin Region, Republic of Sakha(Yakutia) and Omsk Region are still in the basic uncoordinated development type, which their internal system changes from the urbanization backwardness to the urbanization and eco-environment balance. Zabaikalsk Territory, Tomsk Region and Republic of Khakasia are still in the basic uncoordinated development type, Republic of Tyva, Kamchatka Territory and Magadan Region change from the serious uncoordinated development type into the basic uncoordinated development type, Republic of Altay, Jewish Autonomous Area and Chukotka Autonomous Area are still in the serious uncoordinated development type, which their internal system is still in the urbanization backwardness type.

## Author Contributions

**Conceptualization:** Nan-Chen Chu.

**Data curation:** Nan-Chen Chu.

**Formal analysis:** Nan-Chen Chu.

**Funding acquisition:** Nan-Chen Chu.

**Investigation:** Ping-Yu Zhang.

**Methodology:** Nan-Chen Chu.

**Project administration:** Nan-Chen Chu.

**Resources:** Xiang-Li Wu.

**Software:** Nan-Chen Chu.

**Supervision:** Nan-Chen Chu.

**Validation:** Nan-Chen Chu.

**Visualization:** Nan-Chen Chu.

**Writing – original draft:** Nan-Chen Chu.

**Writing – review & editing:** Nan-Chen Chu.

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
