## [Decision Letter · Decision Letter 0]

28 Feb 2022

PONE-D-22-02247Spatiotemporal evolution characteristics of coordinated development of urbanization and ecological environment in eastern RussiaPLOS ONE

Dear Dr. Chu,

Thank you for submitting your manuscript to PLOS ONE. After careful consideration, we feel that it has merit but does not fully meet PLOS ONE’s publication criteria as it currently stands. Therefore, we invite you to submit a revised version of the manuscript that addresses the points raised during the review process.

Specifically, the reviewers have pointed to language related issues, improving the literature review section as well as the discussions section to include comparisons of the findings of the study with those of the literature. There are some other points that need to be revised and these are detailed in reviewers' reports. 

We look forward to receiving your revised manuscript.

Kind regards,

Eda Ustaoglu, PhD

Academic Editor

PLOS ONE

Journal Requirements:

[NO authors have competing interests]. 

6. We note that Figures 1, 5 and 7 in your submission contain map/satellite images which may be copyrighted. All PLOS content is published under the Creative Commons Attribution License (CC BY 4.0), which means that the manuscript, images, and Supporting Information files will be freely available online, and any third party is permitted to access, download, copy, distribute, and use these materials in any way, even commercially, with proper attribution. For these reasons, we cannot publish previously copyrighted maps or satellite images created using proprietary data, such as Google software (Google Maps, Street View, and Earth). For more information, see our copyright guidelines: http://journals.plos.org/plosone/s/licenses-and-copyright.

a) You may seek permission from the original copyright holder of Figures 1, 5 and 7 to publish the content specifically under the CC BY 4.0 license.  

Reviewers' comments:

Reviewer's Responses to Questions

**Comments to the Author**

1. Is the manuscript technically sound, and do the data support the conclusions?

Reviewer #1: Yes

Reviewer #2: Yes

Reviewer #3: Yes

2. Has the statistical analysis been performed appropriately and rigorously? 

Reviewer #1: Yes

Reviewer #2: Yes

Reviewer #3: Yes

3. Have the authors made all data underlying the findings in their manuscript fully available?

Reviewer #1: Yes

Reviewer #2: Yes

Reviewer #3: No

4. Is the manuscript presented in an intelligible fashion and written in standard English?

Reviewer #1: Yes

Reviewer #2: Yes

Reviewer #3: Yes

5. Review Comments to the Author

Reviewer #1: Combining with the Population-Economic-Sociology, Pressure-State-Response models and coupling coordination model, this paper studies the urbanization development level, eco-environment development level, and their coupling coordinated development degree during 2005-2018 in the eastern Russia. The perspectives of the 3D global trend and 2D plane analysis have some characteristics in this paper. Besides, this paper suggests policies and strategies that can boost the growth and development of the urbanization and the eco-environment in the Sino-Russian border areas. Under the background of “the Belt and Road” initiative, it has unique theoretical innovation and academic value. It could provide reference for regional development planning, economic optimization, energy and resource development and infrastructure construction in the adjacent areas of China and Russia in the future. This paper has a complete framework and rigorous method. In particular, this paper selects the eastern region of Russia as the research area, which has some new ideas and characteristics. This paper has reached the level of publication. However, this paper has a long introductory content, which needs to be further simplified. The introduction only points out the current national policy background and the situation of urbanization and ecological environment in eastern Russia, and there are too many background analyses.

Reviewer #2: The coordinated development of urbanization and ecological environment is a scientific issue for urban sustainable development. Based on the index system constructed by the coupling coordination degree model, this paper comprehensively evaluates the coordination degree of urbanization and ecological environment in eastern Russia. Under the background of “the Belt and Road” and “the economic corridor of China, Mongolia and Russia” initiatives, it is of great significance to study the temporal and spatial evolution characteristics of the coordinated development between the urbanization and ecological environment in eastern Russia. The perspectives of the 3D global trend and 2D plane analysis have some innovative significance. This paper could provide policy implications for determining the cooperation direction of border trade, transportation facilities, border tourism, border cooperation zone and ecological environment protection of China and Russia in the future. In terms of the overall level of this paper, this paper has its own characteristics and innovativeness. However, in terms of the index systems, the evaluation of urbanization and the evaluation of ecological environment are essentially two evaluations of this research. It is suggested to distinguish them in Table 1. Because this is not a set of evaluation index system, but two evaluation index systems.

Reviewer #3: Current paper studied the urbanization development level, eco-environment development level, and their coupling coordinated development degree during 2005-2018 in the eastern Russia from the perspectives of the 3D global trend and 2D plane analysis.

The content of the paper is good, but very technical and it is very difficult to follow what authors are trying to say

Overall significant work is needed on the language of the paper.

There are loose sentences and there are many sentences that start with “And”. Authors need to work on bringing them in proper language.

The content is good, but the style of writing is very technical and I feel it may not appeal a wider readership.

Also there are only limited references from past studies to compare and discussion section does not provide any comparison with earlier studies.

Currently introduction is lengthy, it will also help to add sub-headings in introduction section to make the reading easier and crispier

Better to keep labels for on all the maps as you have done for Fig 1

Certain terms such urbanization lagging need better wording and need to be defined in the paper properly

Discussion section does not provide the results of this study with any earlier study, need to add comparison and contrast for validation of the results

Conclusions are not properly formulated, need to work on the language to bring out exactly what are those 2-3 concluding points from this study

Some of the comments are as follows:

Line 13 – passive, instead of this paper studied-In this paper we studied

Line 55, 58, 78…., 653, 657, 662 – sentences starting with And

Line 72, 112 – research instead of researches

Line 75 – Shang analysed.. pls mention citation for Shang

Line 80 – mishchuk citation needed, pls check citation style for the journal

Line 89 – in the new era… need to specify approximate years

6. PLOS authors have the option to publish the peer review history of their article (what does this mean?). If published, this will include your full peer review and any attached files.

Reviewer #1: No

Reviewer #2: No

Reviewer #3: No

---

## [Decision Letter · Decision Letter 1]

6 Apr 2022

Spatiotemporal evolution characteristics of coordinated development of urbanization and ecological environment in eastern Russia

PONE-D-22-02247R1

Dear Dr. Chu,

We’re pleased to inform you that your manuscript has been judged scientifically suitable for publication and will be formally accepted for publication once it meets all outstanding technical requirements.

Kind regards,

Eda Ustaoglu, PhD

Academic Editor

PLOS ONE

Additional Editor Comments (optional):

Reviewers' comments:

Reviewer's Responses to Questions

**Comments to the Author**

1. If the authors have adequately addressed your comments raised in a previous round of review and you feel that this manuscript is now acceptable for publication, you may indicate that here to bypass the “Comments to the Author” section, enter your conflict of interest statement in the “Confidential to Editor” section, and submit your "Accept" recommendation.

Reviewer #1: All comments have been addressed

Reviewer #2: All comments have been addressed

2. Is the manuscript technically sound, and do the data support the conclusions?

Reviewer #1: Yes

Reviewer #2: Yes

3. Has the statistical analysis been performed appropriately and rigorously? 

Reviewer #1: Yes

Reviewer #2: Yes

4. Have the authors made all data underlying the findings in their manuscript fully available?

Reviewer #1: Yes

Reviewer #2: Yes

5. Is the manuscript presented in an intelligible fashion and written in standard English?

Reviewer #1: Yes

Reviewer #2: Yes

6. Review Comments to the Author

Reviewer #1: (No Response)

Reviewer #2: An interesting and valuable study paper, I do not have much too many comments about the studies. The paper is suitable for publication.

7. PLOS authors have the option to publish the peer review history of their article (what does this mean?). If published, this will include your full peer review and any attached files.

Reviewer #1: No

Reviewer #2: No

---

## [Editor Report · Acceptance letter]

23 Jun 2022

PONE-D-22-02247R1 

Spatiotemporal evolution characteristics of coordinated development of urbanization and ecological environment in eastern Russia—Perspectives from the 3D global trend and 2D plane analysis 

Dear Dr. Chu:

I'm pleased to inform you that your manuscript has been deemed suitable for publication in PLOS ONE. Congratulations! Your manuscript is now with our production department. 

Kind regards, 

on behalf of

Dr. Eda Ustaoglu 

Academic Editor

PLOS ONE